# RA-DIT: Retrieval-Augmented Dual Instruction Tuning

**Xi Victoria Lin**\*   **Xilun Chen**\*   **Mingda Chen**\*
**Weijia Shi**   **Maria Lomeli**   **Rich James**   **Pedro Rodriguez**   **Jacob Kahn**
**Gergely Szilvasy**   **Mike Lewis**   **Luke Zettlemoyer**   **Scott Yih**

FAIR at Meta
{victorialin,xilun,mingdachen,scottyih}@meta.com

## Abstract

Retrieval-augmented language models (RALMs) improve performance by accessing long-tail and up-to-date knowledge from external data stores, but are challenging to build. Existing approaches require either expensive retrieval-specific modifications to LM pre-training or use post-hoc integration of the data store that leads to suboptimal performance. We introduce **R**etrieval-**A**ugmented **D**ual **I**nstruction **T**uning (RA-DIT), a lightweight fine-tuning methodology that provides a third option by retrofitting any LLM with retrieval capabilities. Our approach operates in two distinct fine-tuning steps: (1) one updates a pre-trained LM to better use retrieved information, while (2) the other updates the retriever to return more relevant results, as preferred by the LM. By fine-tuning over tasks that require both knowledge utilization and contextual awareness, we demonstrate that each stage yields significant performance improvements, and using both leads to additional gains. Our best model, RA-DIT 65B, achieves state-of-the-art performance across a range of knowledge-intensive zero- and few-shot learning benchmarks, significantly outperforming existing in-context RALM approaches by up to +8.9% in 0-shot setting and +1.4% in 5-shot setting on average.

## 1 Introduction

Large language models (LLMs) excel as zero- and few-shot learners across various tasks (Brown et al., 2020; Chowdhery et al., 2022; Touvron et al., 2023a;b; Anil et al., 2023; OpenAI, 2023). However, because knowledge is represented only in the model parameters, they struggle to capture long-tail knowledge (Tirumala et al., 2022; Sun et al., 2023) and require substantial resources to be kept up-to-date (Miller, 2023). Retrieval-Augmented Language Modeling (RALM) integrates LLMs with non-parametric information retrieval to overcome these limitations (Guu et al., 2020; Borgeaud et al., 2022; Izacard et al., 2022b; Shi et al., 2023b; Ram et al., 2023). By explicitly decoupling knowledge retrieval with the backbone language model, such architectures have exhibited superior performance on knowledge intensive tasks such as open-domain question answering (Lewis et al., 2020; Izacard et al., 2022b) and live chat interactions (Liu, 2022).

Existing RALM architectures focus on two high-level challenges: (i) enhancing the LLM's capability to incorporate retrieved knowledge (Lewis et al., 2020; Izacard et al., 2022b) and (ii) refining the retrieval component to return more relevant content (Shi et al., 2023b; Izacard et al., 2022b). Previous work have also introduced retrieval capabilities at different stages of the model training process. REALM (Guu et al., 2020) and RETRO (Borgeaud et al., 2022) opt for *end-to-end pre-training*, incorporating the retrieval component from the outset. Atlas (Izacard et al., 2022b) builds upon the T5 language model (Raffel et al., 2020), and *continuosly pre-trains* the framework over unsupervised text. REPLUG (Shi et al., 2023b) and In-Context RALM (Ram et al., 2023) combine *off-the-shelf* LLMs with general-purpose retrievers, showing that these two components can be effectively fused through the emergent in-context learning capbabilities of LLMs. However, extensive pre-training of such architectures is expensive, and the off-the-shelf fusion approach also has limitations, particularly as the LLMs are not inherently trained to incorporate retrieved content.

---

\*Equal contribution

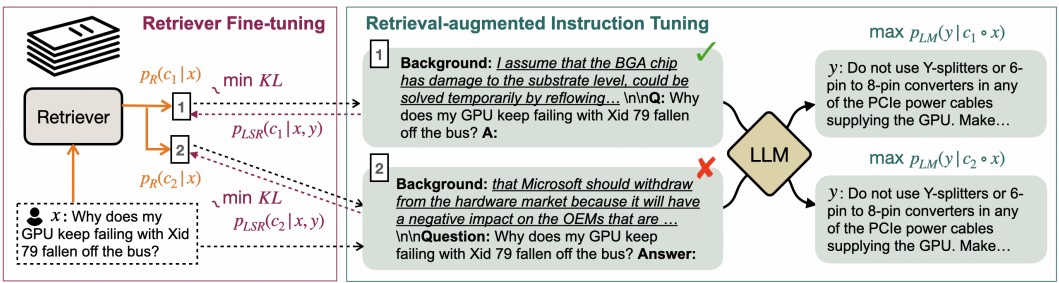

Figure 1: The RA-DIT approach separately fine-tunes the LLM and the retriever. For a given example, the LM-ft component updates the LLM to maximize the likelihood of the correct answer given the retrieval-augmented instructions (§2.3); the R-ft component updates the retriever to minimize the KL-Divergence between the retriever score distribution and the LLM preference (§2.4)

In this work, we show lightweight instruction tuning (Chung et al., 2022b; Iyer et al., 2022; Zhou et al., 2023) alone can significantly boost the performance of RALMs, especially in knowledge intensive scenarios. We propose **R**etrieval-**A**ugmented **D**ual **I**nstruction **T**uning (RA-DIT), an approach that retrofits any LLM with retrieval capabilities via fine-tuning over a set of tasks selected to cultivate knowledge utilization and contextual awareness in the language model predictions. We initialize the framework using pre-trained LLAMA (Touvron et al., 2023a) and a state-of-the-art dual-encoder based dense retriever, DRAGON+ (Lin et al., 2023). Following Shi et al. (2023b), we retrieve relevant text chunks based on the language model prompt. Each retrieved chunk is prepended to the prompt, and the predictions from multiple chunks are computed in parallel and ensembled to produce the final output.

We perform instruction-tuning in two separate steps. For *language model fine-tuning* (LM-ft), we adopt the label-loss objective (Chung et al., 2022b; Iyer et al., 2022) and augment each fine-tuning prompt with a retrieved "background" field prepended to the instructions (Figure 1). We also leverage the design of existing NLP tasks and populate this field with the ground truth context for tasks such as reading comprehension and summarization. By incorporating the background text during fine-tuning, we guide the LLM to optimally utilize the retrieved information and ignore distracting content (Shi et al., 2023a). For *retriever fine-tuning* (R-ft), we update the query encoder using a generalized *LM-Supervised Retrieval* (LSR, Shi et al., 2023b) training objective computed over a combination of supervised tasks and unsupervised text completion. This way we enable the retriever to yield more contextually relevant results, aligned with the preferences of the LLM.

We demonstrate that each fine-tuning step offers significant performance gains, and that the fine-tuned LLM and retriever can be combined to achieve further improvements. Our largest model, RA-DIT 65B, attains state-of-the-art performance in zero- and few-shot settings on knowledge intensive benchmarks, notably surpassing the un-tuned in-context RALM approach on datasets including MMLU (Hendrycks et al., 2021a) (+8.2% 0-shot; +0.7% 5-shot) and Natural Questions (Kwiatkowski et al., 2019) (+22% 0-shot; +3.8% 5-shot). In addition, RA-DIT 65B also substantially outperforms ATLAS 11B on 8 knowledge-intensive tasks (+7.2% on average in the 64-shot fine-tuning setting). This suggests that language models and retrievers, when optimized independently and then fused through instruction-tuning, can compete effectively with RALMs that have undergone extensive continuous pre-training. We further conduct a comprehensive model analysis, showing the effectiveness of our approach across LLMs of varying sizes, as well as evaluating the influence of different fine-tuning strategies and retriever configurations.[1]

---

[1]We release the scripts for indexing Common Crawl data and generating our fine-tuning and inference prompts at: `https://github.com/facebookresearch/RA-DIT`.

## 2 METHOD

### 2.1 ARCHITECTURE

**Language Model**   We focus on retrieval-augmenting pre-trained auto-regressive language models (Brown et al., 2020). In particular, we use LLAMA (Touvron et al., 2023a), a family of open-sourced language models pre-trained on trillions of tokens.

**Retriever**   We adopt a dual-encoder based retriever architecture, since it can be easily fine-tuned and is efficient at the inference stage (Lewis et al., 2020; Izacard et al., 2022b; Shi et al., 2023b). Given a corpus $\mathcal{C}$ and a query $q$, the document encoder maps each *text chunk* $c \in \mathcal{C}$ to an embedding $\mathbf{E}_d(c)$ and the query encoder maps $q$ to an embedding $\mathbf{E}_q(q)$. The top-$k$ relevant text chunks for $q$ are retrieved based on the query-document embedding similarity, which is often computed via dot product:

$$s(q,c) = \mathbf{E}_q(q) \cdot \mathbf{E}_d(c). \tag{1}$$

We initialize the retriever using DRAGON+ (Lin et al., 2023), a state-of-the-art dual-encoder model trained with a contrastive learning objective and large-scale data augmentation.

**Parallel In-Context Retrieval-Augmentation**   Following Shi et al. (2023b), for a given language model prompt $x$, we retrieve the top-$k$ relevant text chunks $\mathcal{C}' \subset \mathcal{C}, |\mathcal{C}'| = k$. To stay within the context window size limit, each retrieved chunk is prepended to the prompt[2], and the language model predictions from multiple augmented prompts are computed in parallel. The final output probability is a mixture of the probability from each augmented prompt weighted by the chunk relevance score

$$p_{LM}(y|x, \mathcal{C}') = \sum_{c \in \mathcal{C}'} p_{LM}(y|c \circ x) \cdot p_R(c|x), \tag{2}$$

where $\circ$ denotes sequence concatenation, and $p_R(c|x) = \frac{\exp s(x,c)}{\sum_{c' \in \mathcal{C}'} \exp s(x,c')}$ are the retriever scores re-normalized among top-$k$ relevant chunks.

### 2.2 FINE-TUNING DATASETS

We choose a set of fine-tuning tasks aimed at boosting the language model's ability to utilize knowledge effectively and improving its contextual awareness in generating predictions. As shown in Table 1, our *language model fine-tuning* datasets ($\mathcal{D}_L$) consists of 20 datasets across 5 distinct categories: dialogue, open-domain QA, reading comprehension[3], summarization and chain-of-thought reasoning. For *retriever fine-tuning* datasets $\mathcal{D}_R$, we opt for the QA datasets in our collection featuring standalone questions, and we additionally include two QA datasets, FreebaseQA (Jiang et al., 2019) and MS-MARCO (Nguyen et al., 2016). The examples of each dataset are serialized for instruction tuning using manually compiled templates (Table 10). For tasks in $\mathcal{D}_L \cap \mathcal{D}_R$, we use the same template for both fine-tuning steps. In addition, we observe that supplementing the instruction-tuning data with unsupervised text leads to additional performance gains for both language model and retriever fine-tuning, and we detail data mixture used in Appendix B.

### 2.3 RETRIEVAL AUGMENTED LANGUAGE MODEL FINE-TUNING

To improve the language model's ability to utilize retrieved information, we fine-tune it on the selected datasets $\mathcal{D}_L$ with in-context retrieval augmentation. Formally, we separate each fine-tuning sequence into an instruction segment ($x$) and an output segment ($y$). For each example $(x_i, y_i) \in$

---

[2]We use a pair of start ("Background:") and end ("\n\n") tokens to demarcate the retrieved segment in the augmented prompt. The complete set of our instruction-tuning templates are shown in Appendix C.

[3]Our reading comprehension (RC) fine-tuning datasets include SQuAD 2.0 (Rajpurkar et al., 2018), which trains the model to determine whether a question can be answered using a given passage, and to provide an answer only when the passage is relevant (otherwise the response is set to "I don't know"). As shown in Appendix F, fine-tuning on this dataset promotes a desirable behavior: the instruction-tuned model tends to respond with "I don't know" when the retriever presents an incorrect passage. We leave further exploring this behavior to improve answer generation as a future work.

Table 1: Our intruction tuning datasets. All datasets are downloaded from Hugging Face (Lhoest et al., 2021), with the exception of those marked with ‡, which are taken from Iyer et al. (2022).

| Task | HF identifier | Dataset name | $\mathcal{D}_L$ | $\mathcal{D}_R$ | #Train |
|---|---|---|---|---|---|
| Dialogue | oasst1 | OpenAssistant Conversations Dataset (Köpf et al., 2023) | ✓ | ✓ | 31,598 |
| Open-Domain QA | commonsense_qa | CommonsenseQA (Talmor et al., 2019) | ✓ | ✓ | 9,741 |
| | math_qa | MathQA (Amini et al., 2019) | ✓ | ✓ | 29,837 |
| | web_questions | Web Questions (Berant et al., 2013) | ✓ | ✓ | 3,778 |
| | wiki_qa | Wiki Question Answering (Yang et al., 2015) | ✓ | ✓ | 20,360 |
| | yahoo_answers_qa | Yahoo! Answers QA | ✓ | | 87,362 |
| | freebase_qa | FreebaseQA (Jiang et al., 2019) | | ✓ | 20,358 |
| | ms_marco* | MS MARCO (Nguyen et al., 2016) | | ✓ | 80,143 |
| Reading Comprehension | coqa | Conversational Question Answering (Reddy et al., 2019) | ✓ | | 108,647 |
| | drop | Discrete Reasoning Over Paragraphs (Dua et al., 2019) | ✓ | | 77,400 |
| | narrativeqa | NarrativeQA (Kočiský et al., 2018) | ✓ | | 32,747 |
| | newsqa | NewsQA (Trischler et al., 2017) | ✓ | | 74,160 |
| | pubmed_qa | PubMedQA (Jin et al., 2019) | ✓ | ✓ | 1,000 |
| | quail | QA for Artificial Intelligence (Rogers et al., 2020) | ✓ | | 10,246 |
| | quarel | QuaRel (Tafjord et al., 2019) | ✓ | ✓ | 1,941 |
| | squad_v2 | SQuAD v2 (Rajpurkar et al., 2018) | ✓ | | 130,319 |
| Summarization | cnn_dailymail | CNN / DailyMail (Hermann et al., 2015) | ✓ | | 287,113 |
| Chain-of-thought Reasoning | aqua_rat‡ | Algebra QA with Rationales (Ling et al., 2017) | ✓ | | 97,467 |
| | ecqa‡ | Explanations for CommonsenseQ (Aggarwal et al., 2021) | ✓ | | 7,598 |
| | gsm8k‡ | Grade School Math 8K (Cobbe et al., 2021) | ✓ | | 7,473 |
| | compeition_math‡ | MATH (Hendrycks et al., 2021b) | ✓ | | 7,500 |
| | strategyqa‡ | StrategyQA (Geva et al., 2021) | ✓ | | 2,290 |

* We only used the question-and-answer pairs in the MS MARCO dataset.

$\mathcal{D}_L$, we retrieve the top-$\tilde{k}$ relevant text chunks $\mathcal{C}_i \subset \mathcal{C}$ based on $x_i$. Mirroring the inference-time handling, for each retrieved chunk $c_{ij} \in \mathcal{C}_i$, we create a separate fine-tuning example by prepending it to the instructions as a background field, resulting in $\tilde{k}$ independent fine-tuning instances per original example: $\{(c_{ij} \circ x_i, y_i) | j = 1 \ldots \tilde{k}\}$.[4]

We fine-tune the language model using the next-token prediction objective and minimize the loss from tokens in the output segment of each instance (Iyer et al., 2022):

$$\mathcal{L}(\mathcal{D}_L) = - \sum_i \sum_j \log p_{LM}(y_i | c_{ij} \circ x_i). \tag{3}$$

Integrating in-context retrieval augmentation during fine-tuning gives a twofold benefit. First, it adapts the LLM to better utilize relevant background knowledge to make a prediction. Secondly, even state-of-the-art retrievers can falter and return inaccurate results. By training the LLM to make correct predictions when a wrong retrieved chunk is given, we enable the LLM to ignore misleading retrieval content and lean into its parametric knowledge in such cases. The efficacy of this fine-tuning strategy is empirically demonstrated in §5.1.

## 2.4 RETRIEVER FINE-TUNING

In addition to fine-tuning the language model with retrieval augmentation, we also fine-tune the retriever to better align its output with the language model. In particular, we adopt a generalized version of LSR (*LM-Supervised Retrieval*, Shi et al., 2023b) training that leverages the language model itself to provide supervision for retriever fine-tuning.

For a training sample $(x, y)$ in the retriever fine-tuning dataset $\mathcal{D}_R$, we define the LSR score for a retrieved chunk $c$ as follows:

$$p_{LSR}(c|x,y) = \frac{\exp\left(p_{LM}(y|c \circ x)/\tau\right)}{\sum_{c' \in \mathcal{C}} \exp\left(p_{LM}(y|c' \circ x)/\tau\right)} \approx \frac{\exp\left(p_{LM}(y|c \circ x)/\tau\right)}{\sum_{c' \in \mathcal{C}'} \exp\left(p_{LM}(y|c' \circ x)/\tau\right)}, \tag{4}$$

where $\tau$ is a temperature hyperparameter, and $\mathcal{C}' \subset \mathcal{C}$ denotes the top-$k$ retrieved chunks for $x$. A higher LSR score indicates that $c$ is more effective at improving the language model's chance of

[4]The exceptions are summarization tasks and RC tasks with context dependent questions (e.g. "when was the writer born?"), where we do not perform retrieval and create the fine-tuning instances using the given background text instead. For RC tasks with self-contained questions, we use the retrieved chunks in addition to the given background text to create fine-tuning instances, resulting in $\tilde{k} + 1$ of them per original example.

predicting the correct answer. The goal of LSR training is for the retriever to assign higher scores to chunks that can improve the LLM's likelihood of generating the correct answer. To achieve this, we minimize the KL-divergence between $p_{LSR}$ and the retriever scores $p_R$ defined in Eq. 2:

$$\mathcal{L}(\mathcal{D}_R) = \mathbb{E}_{(x,y) \in \mathcal{D}_R} KL\big(p_R(c|x) \parallel p_{LSR}(c|x,y)\big) \tag{5}$$

In practice, we only update the query encoder of the retriever, as fine-tuning both encoders hurts the performance (§5.1). While previous work (Shi et al., 2023b) relies solely on unlabeled texts (denoted as *corpus data*) for LSR training, we show that LSR can be generalized to incorporate the multi-task instruction data introduced in §2.2 (denoted as *MTI data*). The MTI data provide direct supervision to the retriever to return relevant information that enhances the language model in various downstream tasks. As shown in §5.1, combining both types of data yields the best results and outperforms using either source alone.

## 3 EXPERIMENT SETUP

### 3.1 RETRIEVER

We initialize the retriever in our framework with DRAGON+ (Lin et al., 2023) and also use it to study various retriever configurations. To build the retrieval corpus, we combine the text chunks from the Dec. 20, 2021 Wikipedia dump released by Izacard et al. (2022b) with additional ones from the 2017-2020 CommonCrawl dumps. We detail the corpus pre-processing and indexing in Appendix A. Our final retrieval data store, with the two data sources combined, contain 399M text chunks with a maximum length of 200 words. In Appendix E.3, we conduct an analysis on the impact of using various subsets of the retrieval corpora, as well as different Wikipedia snapshots. We obtain the retrieval queries used for our fine-tuning and evaluation tasks using manually[5] constructed templates (Table 10 and 12).

### 3.2 BASELINES

We focus on comparing our approach to the base LLAMA models (Touvron et al., 2023a) and RE-PLUG (Shi et al., 2023b), a state-of-the-art approach that integrates off-the-shelf LLMs and retrievers, in the zero-shot and in-context few-shot learning settings. We instantiate REPLUG using LLAMA and DRAGON+. In addition, we also compare RA-DIT to ATLAS (Izacard et al., 2022b) in a 64-shot fine-tuning setting (§4).

### 3.3 EVALUATION

We primarily conduct evaluation on knowledge-intensive tasks that are not included in our fine-tuning datasets, including MMLU (Hendrycks et al., 2021a), Natural Questions (NQ; Kwiatkowski et al., 2019), TriviaQA (TQA; Joshi et al., 2017), and a subset[6] of the tasks in the KILT benchmark (Petroni et al., 2021). We use the development split of the KILT subset excluding ELI5 to determine fine-tuning hyperparameters (Appendix B). This enables us to report genuine few-shot evaluation results for 4 out of the 10 evaluation tasks. For the remaining tasks, we report few-shot results assuming access to in-domain development data. In addition, we also evaluate the models on commonsense reasoning tasks to measure the impact of the proposed approach on the LLM's parametric knowledge and reasoning capabilities. Details of our evaluation datasets, including the evaluation metrics, template and the scoring functions used, can be found in in Appendix D.

## 4 MAIN RESULTS

**Knowledge-Intensive Tasks** We report the main results in Table 2. In particular, RA-DIT is compared to LLAMA (Touvron et al., 2023a) as well as REPLUG (Shi et al., 2023b), in both 0-shot and

---

[5] We leave automatically generating task-specific retrieval queries to future work.

[6] The subset consists of seven tasks: HotpotQA (Yang et al., 2018), FEVER (Thorne et al., 2018), AIDA CoNLL-YAGO (Hoffart et al., 2011), Zero-Shot RE (Levy et al., 2017), T-REx (Elsahar et al., 2018), Wizard of Wikipedia (Dinan et al., 2019) and ELI5 (Fan et al., 2019).

Table 2: Main results: Performance on knowledge intensive tasks (test sets).

| | MMLU | NQ | TQA | ELI5 | HoPo | FEV | AIDA | zsRE | T-REx | WoW | Avg$^\diamond$ | Avg |
|---|---|---|---|---|---|---|---|---|---|---|---|---|
| *0-shot* | | | | | | | | | | | | |
| LLAMA 65B | 51.2 | 5.2 | 55.8 | 19.5 | 12.5 | 59.3 | 0.6 | 6.7 | 1.3 | 15.6 | 32.9 | 22.8 |
| LLAMA 65B REPLUG | 59.7 | 28.8 | 72.6 | 19.1 | 32.0 | 73.3 | 41.8 | 50.8 | 36.3 | 16.1 | 45.1 | 43.1 |
| RA-DIT 65B | **64.6** | **35.2** | **75.4** | **21.2** | **39.7** | **80.7** | **45.1** | **73.7** | **53.1** | **16.4** | **49.1** | **50.5** |
| *5-shot in-context* | | | | | | | | | | | | |
| LLAMA 65B | 63.4 | 31.6 | 71.8 | 22.1 | 22.6 | 81.5 | 48.2 | 39.4 | 52.1 | **17.4** | 47.2 | 45.0 |
| LLAMA 65B REPLUG | 64.4 | 42.3 | 74.9 | 22.8 | **41.1** | 89.4 | 46.4 | 60.4 | **68.9** | 16.8 | 51.1 | 52.7 |
| RA-DIT 65B | **64.9** | **43.9** | **75.1** | **23.2** | 40.7 | **90.7** | **55.8** | **72.4** | 68.4 | 17.3 | **51.8** | **55.2** |

| *64-shot fine-tuned* | NQ | TQA | HoPo | FEV | AIDA | zsRE | T-REx | WoW | Avg |
|---|---|---|---|---|---|---|---|---|---|
| ATLAS[†] | 42.4 | **74.5** | 34.7 | **87.1** | 66.5 | 74.9 | 58.9 | 15.5 | 56.8 |
| RA-DIT 65B | **43.5** | 72.8 | **36.6** | 86.9 | **80.5** | **78.1** | **72.8** | **15.7** | **60.9** |

$^\diamond$ Average of MMLU, NQ, TQA, and ELI5.

[†] ATLAS conducts 64-shot fine-tuning for each individual task and evaluates task-specific models individually. For RA-DIT, we perform multi-task fine-tuning using 64-shot examples from each task combined, and report the performance of a unified model across tasks.

Table 3: Performance on commonsense reasoning tasks (dev sets) without retrieval augmentation.

| *0-shot* | BoolQ | PIQA | SIQA | HellaSwag | WinoGrande | ARC-E | ARC-C | OBQA | Avg |
|---|---|---|---|---|---|---|---|---|---|
| LLAMA 65B | 85.3 | 82.8 | 52.3 | 84.2 | 77.0 | 78.9 | 56.0 | **60.2** | 72.1 |
| RA-DIT 65B | **86.7** | **83.7** | **57.9** | **85.1** | **79.8** | **83.7** | **60.5** | 58.8 | **74.5** |

5-shot settings. We first observe that REPLUG works much better than the base LLAMA 65B, confirming the benefits of RALMs on knowledge-intensive tasks. Furthermore, RA-DIT significantly outperforms REPLUG (+8.9% in 0-shot and +1.4% in 5-shot on average over MMLU, NQ, TQA and ELI5) and achieves the best performance on most datasets. This corroborates our claim that combining off-the-shelf LLMs and retrievers is sub-optimal, and our dual instruction tuning approach is an effective way of retrofitting LLMs with retrieval capabilities.[7]

We also compare with ATLAS, a state-of-the-art encoder-decoder based RALM that jointly pre-trains the language model and the retriever. Here we adopt a 64-shot setting similar to Izacard et al. (2022b) with the following differences. While ATLAS conducts 64-shot fine-tuning for each individual task and reports the performance of task-specific models, we continuously fine-tune the RA-DIT checkpoint using the 64-shot examples from all tasks combined, and report the performance of a single model across tasks. As shown in Table 2, despite using a single model, RA-DIT outperforms ATLAS by an average of 4.1 points, achieving higher performance on 6 out of the 8 datasets.

**Commonsense Reasoning** We benchmark RA-DIT 65B on a set of commonsense reasoning tasks to evaluate the impact of retrieval-augmented instruction tuning on the LLM's parametric knowledge and reasoning capabilities. We hence do not perform retrieval augmentation in this experiment. As shown in Table 3, RA-DIT demonstrates improvements over the base LLAMA models on 7 out of 8 evaluation datasets, indicating that the parametric knowledge and reasoning capabilities of the LLM component are in general preserved. As discussed in Appendix F, maintaining the parametric knowledge in the LLM component is vital as a safety net when the retriever makes mistakes.

# 5 ANALYSIS

## 5.1 FINE-TUNING STRATEGIES

**Language Model Fine-tuning** We compare LLAMA instruction-tuned with retrieval-augmentation (RA-IT 65B) to the base language model, as well as LLAMA that is instruction-tuned

---

[7]We report lower 0-shot performance for LLAMA 65B on NQ and TQA in comparison to Touvron et al. (2023a). By examining the model generation, we think Touvron et al. (2023a) reported the ratio of responses that contain the ground truth answer string in the 0-shot setting, while we report exact match.

Table 4: Ablation of language model fine-tuning strategies. All rows report dev set performance.

| 0 / 5-shot | HoPo | FEV | AIDA | zsRE | T-REx | WoW | Avg |
|---|---|---|---|---|---|---|---|
| LLAMA 65B | 12.5 / 23.8 | 59.6 / 83.7 | 0.9 / 64.1 | 9.7 / 36.0 | 1.2 / 52.3 | 15.7 / 17.4 | 16.6 / **46.2** |
| IT 65B | 20.0 / 30.0 | 67.8 / 83.2 | 8.9 / 58.5 | 19.0 / 35.4 | 17.3 / 53.5 | 16.4 / 16.5 | 24.9 / **46.2** |
| RA-IT 65B | 26.8 / 29.9 | 65.2 / 84.8 | 10.7 / 52.9 | 30.9 / 35.2 | 24.1 / 52.9 | 16.5 / 16.5 | **29.0** / 45.4 |
| *top-1 chunk* | | | | | | | |
| LLAMA 65B + DRAGON+ | 25.8 / 39.4 | 72.8 / 89.8 | 39.1 / 50.7 | 48.8 / 59.6 | 31.4 / 69.1 | 15.8 / 17.1 | 39.0 / **54.3** |
| IT 65B + DRAGON+ | 33.3 / 38.8 | 84.0 / 90.1 | 43.9 / 50.3 | 56.8 / 58.2 | 44.7 / 66.4 | 15.7 / 15.6 | 46.4 / 53.2 |
| RA-IT 65B + DRAGON+ | 37.6 / 39.1 | 81.0 / 90.4 | 41.6 / 52.3 | 59.6 / 57.9 | 49.6 / 65.8 | 16.6 / 16.6 | **47.7** / 53.7 |
| *top-3 chunks* | | | | | | | |
| LLAMA 65B + DRAGON+ | 29.6 / 40.8 | 74.9 / 90.3 | 43.1 / 52.8 | 55.9 / 62.9 | 37.2 / 70.8 | 16.0 / 17.2 | 42.8 / **55.8** |
| IT 65B + DRAGON+ | 35.2 / 40.0 | 85.7 / 91.2 | 49.7 / 52.3 | 56.2 / 61.9 | 45.9 / 68.6 | 15.6 / 15.6 | 48.1 / 54.9 |
| RA-IT 65B + DRAGON+ | 39.9 / 40.6 | 82.4 / 91.7 | 45.2 / 53.4 | 63.4 / 61.3 | 52.8 / 67.6 | 16.6 / 16.7 | **50.1** / 55.2 |
| *top-10 chunks* | | | | | | | |
| LLAMA 65B + DRAGON+ | 31.0 / 41.6 | 75.4 / 90.8 | 44.8 / 54.0 | 58.6 / 63.7 | 40.2 / 71.9 | 16.0 / 17.8 | 44.3 / **56.6** |
| IT 65B + DRAGON+ | 33.9 / 40.6 | 87.0 / 91.8 | 50.5 / 53.8 | 53.9 / 62.5 | 45.7 / 69.4 | 15.6 / 15.7 | 47.8 / 55.6 |
| RA-IT 65B + DRAGON+ | 40.0 / 41.2 | 82.8 / 92.1 | 47.2 / 53.5 | 65.0 / 62.3 | 54.3 / 69.0 | 16.5 / 16.6 | **51.0** / 55.8 |

Table 5: Ablation of retriever fine-tuning strategies. All rows use the LLAMA 65B model and report 5-shot performance on the dev sets.

| 5-shot | MMLU | NQ | TQA | HoPo | FEV | AIDA | zsRE | T-REx | WoW | Avg$^{\diamond}$ | Avg |
|---|---|---|---|---|---|---|---|---|---|---|---|
| DRAGON+ | 62.6 | 41.8 | 72.9 | 41.5 | 90.6 | 54.1 | 63.7 | 72.1 | 17.5 | 56.6 | 57.4 |
| MTL instruction tuning data | 61.1 | 43.6 | 74.0 | 36.5 | 91.4 | 64.6 | 56.7 | 72.1 | 17.1 | 56.4 | 57.5 |
| corpus data (FT both encoders) | 61.7 | 43.2 | 73.8 | 37.5 | 88.2 | 69.8 | 53.5 | 57.2 | 17.5 | 54.0 | 55.8 |
| corpus data | 62.9 | 43.0 | 74.3 | 41.1 | 91.6 | 54.4 | 63.4 | 71.8 | 17.4 | 56.6 | 57.8 |
| 95% corpus + 5% MTL data | 63.0 | 42.1 | 74.9 | 41.2 | 91.6 | 54.9 | 65.2 | 71.6 | 17.5 | **57.0** | **58.0** |

$^{\diamond}$ Average over the 6 KILT development tasks.

conventionally[8] (IT 65B) on the same set of tasks. We evaluate all models with in-context retrieval augmentation using the DRAGON+ retriever, adjusting the number of retrieved chunks to 0, 1 or 10. As shown in Table 4, while both instruction tuning methods substantially enhance the 0-shot performance, they offers marginal improvements or even hurt the model performance in the 5-shot setting for most tasks except for HotpotQA[9]. When in-context retrieval-augmentation is applied, all models show substantial gains in both settings, even when limited to the top-1 chunk. The model performance consistently improves as we include more retrieved chunks. In the 0-shot setting with top-10 retrieved chunks, the RA-IT 65B model outperforms the IT 65B model by a large margin (51.0% vs. 47.7%). Under this setting, we observe that retrieval-augmented instruction tuning significantly enhances the LLM's ability to integrate information from the retrieved text chunks. The model is able to extract the correct answers from relevant chunks with greater confidence, while effectively leaning on its parametric knowledge for prediction when an irrelevant text chunk is present (Appendix F). In Appendix E.1, we also discuss the performance of RA-IT models when applied to smaller LLAMA models (7B and 13B), showing that it offers even larger performance boost in those cases.

**Retriever Fine-tuning** In Table 5, we study different retriever fine-tuning strategies. As mentioned in §2.4, we explore two types of retriever fine-tuning data, the *multi-task instruction (MTI) data* and the *corpus data*. We observe that fine-tuning the retriever with the corpus data alone improves over the base DRAGON+ model by an average of 0.4 points, whereas fine-tuning using only the MTI data improves by a smaller margin of 0.1 points. While fine-tuning with the MTI data yields good performance on certain datasets such as NQ (possibly due to its similarity to the MTI data), fine-tuning with the corpus data appears to generalize better and leads to stronger overall performance. Furthermore, we experiment with fine-tuning using both the MTI and corpus data. Table 5

---

[8]Since our instruction tuning datasets include reading comprehension and summarization, the IT models are also exposed to problem types that depend on background knowledge.

[9]This observation aligns with the findings from previous instruction-tuning literature (Iyer et al., 2022). HotpotQA is an exception likely because it is from a task category covered in our instruction-tuning data.

shows that fine-tuning with "95% corpus data + 5% MTI data" achieves the best accuracy across all models, outperforming the non-finetuned baseline by 0.6 points on average.[10]

Finally, we also compare jointly fine-tuning both the query and document encoders with only fine-tuning the query encoder while freezing the document encoder. Table 5 shows this experiment conducted using the corpus data, where freezing the document encoder produces significantly better performance. As a result, we only fine-tune the query encoder in this work.

## 5.2 DUAL INSTRUCTION TUNING ABLATION

Table 6: The impact of LM and Retriever fine-tuning in our RA-DIT method, comparing the RE-PLUG baseline, LM-ft only, R-ft only, and RA-DIT. 5-shot dev set performance is reported.

| *5-shot* | MMLU | NQ | TQA | ELI5 | HoPo | FEV | AIDA | zsRE | T-REx | WoW | Avg |
|---|---|---|---|---|---|---|---|---|---|---|---|
| LLAMA 65B + DRAGON+ | 61.7 | 41.7 | 73.0 | 22.1 | 41.6 | 90.8 | 54.0 | 63.7 | 71.9 | 17.2 | 53.8 |
| LLAMA 65B + FTed DRAGON+ | 63.0 | 42.2 | 74.9 | 22.2 | 41.4 | 91.6 | 54.9 | 65.2 | 71.4 | 17.4 | 54.4 |
| RIT 65B + DRAGON+ | 64.8 | 42.8 | 73.1 | 23.6 | 41.2 | 92.1 | 53.5 | 62.3 | 69.0 | 16.6 | 53.9 |
| RIT 65B + FTed DRAGON+ | 64.3 | 43.8 | 75.0 | 23.3 | 42.0 | 92.3 | 52.8 | 65.2 | 70.1 | 17.3 | **54.6** |

We isolate the impact of the language model fine-tuning from retriever fine-tuning in our RA-DIT method, and illustrate the benefit of each. [11] According to Table 6, both LM-ft and R-ft are beneficial when used alone, and outperform the REPLUG using LLAMA 65B and the DRAGON+ retriever. On the other hand, the most gain can be achieved when combining LM-ft and R-ft in our RA-DIT method, which outperforms the REPLUG baseline by 0.8 points on average. In our preliminary experiments, we also attempted iterative dual instruction tuning by fine-tuning the retriever using LSR scores from the RA-IT LM or conduct the RA-IT step using passages returned by the fine-tuned retriever, for one or two such iterations, but did not observe further gains. We leave the exploration of multi-step RA-DIT to future work.

## 5.3 RETRIEVER SETTINGS

Table 7: Retriever settings: We report 5-shot dev set performance using LLAMA 65B and various retrievers in the REPLUG setting.

| *5-shot* | MMLU | NQ | TQA | HoPo | FEV | AIDA | zsRE | T-REx | WoW | ELI5 | Avg |
|---|---|---|---|---|---|---|---|---|---|---|---|
| LLAMA 65B | 61.3 | 30.9 | 70.6 | 23.8 | 83.7 | 50.2 | 36.0 | 52.3 | 17.4 | 23.4 | 45.0 |
| *Retriever ablation using* LLAMA *65B and the 399M CC + Wiki corpus* | | | | | | | | | | | |
| Contriever | 59.3 | 41.2 | 73.0 | 32.4 | 88.1 | 45.0 | 40.8 | 56.1 | 17.2 | 21.6 | 47.5 |
| Contriever-msmarco | 62.0 | 42.1 | 74.1 | 38.7 | 89.3 | 49.3 | 60.2 | 62.9 | 17.4 | 21.8 | 51.8 |
| DRAGON+ | 61.7 | 41.7 | 73.0 | 40.8 | 90.8 | 48.8 | 63.7 | 71.9 | 17.8 | 23.8 | 53.4 |

We study the impact of various retriever choices in our framework. We use LLAMA 65B as the language model and combine it with different retrievers. Table 7 first compares DRAGON+ (Lin et al., 2023) with other state-of-the-art retrievers such as Contriever (Izacard et al., 2022a). All retrieval-augmented models substantially improve over the LLAMA baseline, and DRAGON+ significantly outperforms both Contriever and Contriever-MSMARCO. We hence adopt DRAGON+ as our base retriever in all experiments.

## 6 RELATED WORK

**Retrieval-Augmented Language Models** RALMs augment LMs with a non-parametric memory to facilitate external knowledge access and provide provenance (Guu et al., 2020; Lewis et al., 2020;

---

[10]In early experiments, we also tested other mixtures and found that using 5% or 10% MTI data worked the best. (They perform similarly to each other.)

[11]Minor performance differences may be observed for the LLAMA 65B + DRAGON+ model in different ablations due to the differences in few-shot example truncation in long prompts. We ensure all rows within each table are comparable.

Borgeaud et al., 2022; Shi et al., 2023b). Previous work have proposed different ways of fusing the LM and the non-parametric component. For example, RETRO (Borgeaud et al., 2022) and FiD (Izacard & Grave, 2021; Hofstätter et al., 2022) leverage separate encoder modules to encode the retrieved content, which are integrated with the backbone LM via cross-attention. A more widely adopted approach directly augments the LM input with the retrieved content (Guu et al., 2020; Lewis et al., 2020; Shi et al., 2023b). This approach yields competitive results with a moderate inference cost increase, as the LM can effectively contextualize the retrieved content and the original prompt through multi-layer self-attention. RA-DIT is grounded in the in-context RA framework for its simplicity and practicality. Instead of performing extensive pre-training (Guu et al., 2020; Borgeaud et al., 2022; Izacard et al., 2022b), we propose a lightweight fine-tuning recipe that primarily utilizes downstream data, and demonstrate improved few-shot generalization of the fine-tuned RALM on knowledge-intensive language tasks.

**Instruction Tuning** Instruction tuning has been proposed to align pre-trained LLMs to follow natural language instructions and avoid extensive prompt engineering (Ouyang et al., 2022; Wei et al., 2022; Chung et al., 2022a; Wang et al., 2022; Iyer et al., 2022). We propose retrieval-augmented instruction tuning (RA-IT) as part of our *dual instruction tuning* framework to improve the LM's ability to leverage retrieved information. Concurrent work has also applied instruction tuning to other RALM architectures. Notably, Wang et al. (2023) fine-tunes the backbone LM in the RETRO architecture while freezing the cross-attention module and the memory encoder. In comparison, RA-DIT fine-tunes both the LM and the retriever while decoupling the fine-tuning processes of the two components.[12] Asai et al. (2023) fine-tunes an LM to adaptively retrieve passages on demand and reflect on the relevancy of the retrieved passages and its generation using special-token markups. The most related work to ours is SAIL (Luo et al., 2023), an approach that fine-tunes the LM with instructions augmented with retrieved content, and examines it on public instruction following datasets (Taori et al., 2023; Chiang et al., 2023) using a moderately sized model (7B parameters). In comparison, RA-DIT conducts parallel retrieval-augmentation for multiple retrieved passages while SAIL concatenates them in the LM context. Furthermore, RA-DIT adopts a holistic view of the RALM architecture by employing a learnable neural retriever and proposing a dual optimization framework. SAIL, in comparison, leans on non-differentiable retrievers such as BM25 and focuses on improving the LM (e.g. it proposes an in-context retrieval selection technique to guide the model focus towards informative content).

**Information Retrieval** Retrieval methods include *sparse retrievers* that does matching over a sparse bag-of-words representation (Robertson & Zaragoza, 2009; Formal et al., 2021), *dense retrievers* that embed queries and documents into a fixed-size dense vector for nearest-neighbor search (Karpukhin et al., 2020; Xiong et al., 2021), and *multi-vector retrievers* which uses multiple vectors as the representation and more complex search algorithms for increased accuracy (Khattab & Zaharia, 2020; Li et al., 2023). We adopt a state-of-the-art dense retriever, DRAGON (Lin et al., 2023), as our base retriever, because of its simplicity, state-of-the-art accuracy, high retrieval efficiency on GPUs, and the ease of further fine-tuning.

## 7 CONCLUSION

In this paper, we propose RA-DIT, a lightweight Retrieval-Augmented Dual Instruction Tuning framework that can effectively retrofit any pre-trained LLM with retrieval capabilities. RA-DIT updates the LLM with *retrieval-augmented instruction tuning* to make better use of retrieved knowledge and ignore irrelevant or distracting information. It also fine-tunes the retriever with supervision from the LLM to retrieve texts that can better help the LLM generate correct outputs. RA-DIT achieves state-of-the-art performance in zero- and few-shot evaluations on knowledge intensive benchmarks, surpassing un-tuned in-context RALM approaches such as REPLUG and compete effectively against methods that require extensive pre-training such as ATLAS.

---

[12] Although the differences in the base LMs, fine-tuning datasets and inference settings make direct comparisons between the two models challenging, RA-DIT 65B compares favorably to InstructRetro 48B (Wang et al., 2023) in zero-shot setting on the shared evaluation datasets.

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

# A  RETRIEVAL CORPUS

We combine the text chunks from the Dec. 20, 2021 Wikipedia dump released by Izacard et al. (2022b) with additional ones from the 2017-2020 CommonCrawl dumps. The Wikipedia dump includes lists and infoboxes in addition to regular articles. The articles are split by section, where long sections are further split into text chunks of equal sizes and contain less than 200 words, leading to a total of 37M text chunks. We randomly sample a subset of articles from the CommonCrawl dumps, and split them into equal-sized text chunks that contain less than 100 white-space-separated words, leading to a total of 362M text chunks.

We use a GPU-based exact $k$-nearest-neighbor search index implementation[13] released by Izacard et al. (2022b).

# B  IMPLEMENTATION DETAILS

**Fine-tuning Dataset Selection**  Prior work (Chung et al., 2022b; Iyer et al., 2022) have demonstrated that jointly fine-tuning the language model on a diverse collection of instruction-based datasets leads to improved model generalization for unseen instructions. We adopt a similar strategy by combining five categories of fine-tuning tasks to enhance the language model's knowledge utilization (dialogue, open-domain QA, chain-of-thought reasoning) and to improve its contextual awareness for prediction generation (reading comprehension, summarization). These categories were selected due to their representativeness of practical knowledge-intensive language tasks.

**Retrieval-augmented LM Fine-tuning**  We use the top-3 retrieved text chunks for a given example (i.e. $\tilde{k} = 3$) to generate the fine-tuning instances. To improve fine-tuning efficiency, we pack multiple examples up to the language model context window limit (2048 tokens). Each example is demarcate by a pair of `<bos>` and `<eos>` tokens, and we adopt the document attention masking (Iyer et al., 2022) such that a token only attends to the previous tokens in the same example. We use a dataset mixture that contains 10% unsupervised text and 5% OASST-1 data. For the remaining datasets, we establish a cap on the number of examples per dataset at $\eta = 7500$ based on the model performance on our development set.[14] We then randomly sample batches in accordance with this adjusted mixture probability.

We fine-tune the 7B, 13B and 65B LLAMA models using 8, 16 and 64 A100 GPUs, respectively. The fine-tuning hyperparameters are detailed in Table 8. Similar to Zhou et al. (2023), we found that the best generalization performance on the dev set can be achieved using a small number of fine-tuning steps. We evaluate the models every 100 steps, and select the best checkpoint based on the average dev set performance over the 6 development KILT tasks shown in Table 11 (early stopping).

Table 8: Hyperparameters for retrieval-augmented LM fine-tuning.

| Model | peak lr | end lr | lr scheduler | warm-up | # steps | early stopping | batch size | model parallel | seq len |
|-------|---------|--------|--------------|---------|---------|----------------|------------|----------------|---------|
| RA-DIT 7B | 1e-5 | 1e-7 | cosine | 200 | 500 | 500 | 64 | 1 | 2048 |
| RA-DIT 13B | 1e-5 | 1e-7 | cosine | 200 | 500 | 400 | 128 | 2 | 2048 |
| RA-DIT 65B | 1e-5 | 1e-7 | cosine | 200 | 500 | 300 | 128 | 8 | 2048 |

**64-shot Eval Task Fine-tuning**  Table 9 summarizes our hyperparameters for 64-shot fine-tuning on the 9 KILT eval tasks shown in Table 12 except for MMLU. Given the small amount of examples used ($64 \times 9 = 576$), we fine-tune for a significantly less number of steps at this stage without using warm-up. We evaluate the model every 50 steps, and select the best checkpoint based on the average dev set performance over the 6 development KILT tasks shown in Table 11.

**Retriever Fine-tuning**  We employ both unsupervised text and downstream tasks for retriever fine-tuning. For the *corpus data*, we randomly sample 900k text chunks from our retrieval corpus to

---

[13] https://github.com/facebookresearch/atlas

[14] We did not thoroughly tune this parameter to avoid overfitting to the development sets.

Table 9: Hyperparameters for 64-shot fine-tuning on the eval tasks.

| Model | peak lr | end lr | lr scheduler | warm-up | # steps | early stopping | batch size | model parallel | seq len |
|---|---|---|---|---|---|---|---|---|---|
| LLAMA 65B | 1e-5 | 1e-6 | linear | 0 | 100 | 100 | 8 | 8 | 2048 |
| RA-DIT 13B | 1e-5 | 1e-6 | linear | 0 | 100 | 50 | 32 | 2 | 2048 |
| RA-DIT 65B | 1e-5 | 1e-6 | linear | 0 | 100 | 50 | 32 | 8 | 2048 |

form a set of self-supervised data, using the first 50 tokens of each chunk as the input $x$ and the last 50 tokens as the ground-truth output $y$. In addition, we leverage the multi-task instruction tuning datasets (MTI data) as shown in Table 1, including 10 open-domain question answering and dialog tasks, with a total of 286k training examples. As discussed in §5.1, we observe that, when used alone, the corpus data works slightly better than the downstream tasks. However, combining both types of fine-tuning data yields the best results and outperforms using either source alone. Therefore, we adopt a mixture of 95% corpus data and 5% downstream tasks for retriever fine-tuning in our final model.

We fine-tune the DRAGON+ retriever on 16 A100 GPUs using the dpr-scale codebase[15]. The retriever is fine-tuned using a learning rate of 1e-5 with 1237 warmup steps (DRAGON default), a per-GPU batch size of 32, and a temperature $\tau = 0.01$, for a single epoch over a combination of 5% *MTI data* and 95% *corpus data*. We adopt the KL-divergence loss as discussed in Section 2.4 using the top-10 retrieved chunks for each example. For simplicity and efficiency, we produce the top-10 retrieved chunks and their LSR scores (Eqn. 4) using LLAMA 65B and DRAGON+, and do not update them during R-ft. Furthermore, as only the query encoder is fine-tuned, there is no need to update the chunk embeddings in the retriever index. Model validation is performed once every 500 steps using the same mean reciprocal rank (MRR) metric as in the original DRAGON paper (Lin et al., 2023), on a combined validation set from the 10-task MTI data.

**Inference** Without further specification, we use the top-10 retrieved text chunks for a given example (i.e. $k = 10$) and ensemble their predictions during inference. For multi-choice tasks, we compute the weighted average probability of each choice items according to Eq. 2 and select the choice with the highest probability. For generation tasks, we perform decoding using each augmented prompt independently, compute the weighted average probability of each unique generated answer, and output the answer with the highest probability.[16] When computing probabilities of output answers, we use several scoring functions: "nll", "nll_char", "nll_token", and "nll_compl". "nll" is the sum of negative log likelihood across all tokens in the sequence. "nll_char" and "nll_token" are "nll" divided by the numbers of characters and subword units in output answers respectively. "nll_compl" selects answers based on the probability divided by the probability of the answer given "Answer:": $\frac{p(y|x)}{p(y|\text{``Answer:''})}$.

## C  FINE-TUNING DATASET TEMPALTES

Table 10 shows the templates we used to serialize our instruction tuning datasets. Following Chung et al. (2022b) and Iyer et al. (2022), we randomize the field markers used during training to avoid overfitting. In pariticular, when serializing a task example, we randomly sample from {"Q:", "Question: ", and ""} for `<inst_s>`, set `<inst_e>` to "\n" and randomly sample from {"A:", "Answer:"} for `<answer_s>`.

## D  EVALUATION DATASETS AND TEMPLATES

Table 11 shows the evaluation datasets used in our experiments. For dev set evaluation, we use a maximum of 2500 randomly sampled examples from the respective official dev sets to reduce the

---

[15]`https://github.com/facebookresearch/dpr-scale`

[16]A more sophisticated implementation of ensembling for generation tasks involves computing a weighted ensemble of the output distribution at every step and then sampling from this distribution. However, we opt for the simpler implementation as it performs reasonably well and allows us to execute inference with fewer GPUs.

Table 10: Instruction template used for our fine-tuning datasets. `<inst_s>`, `<inst_e>` and `<answer_s>` are special markers denoting the start and the end of a field.

| Category | Instruction Tuning Template | Query Template |
|---|---|---|
| Dialogue | Background: {retrieved passage}\n\nQ: {turn$_1$} A: {turn$_2$} Q: {turn$_3$} A: ... | {turn$_1$} {turn$_2$} {turn$_3$} ... |
| Open-domain QA | Background: {retrieved passage}\n\n`<inst_s>` {question} `<inst_e>` `<answer_s>` {answer} | {question} |
| Reading Comprehension | Background: {context}\n\n`<inst_s>` {question} `<inst_e>` `<answer_s>` {answer} | {question} |
| Summarization | Background: {context}\n\nSummarize this article: `<inst_e>` `<answer_s>` {summary} | |
| Chain-of-thought Reasoning | Background: {retrieved passage}\n\n`<inst_s>` {instructions} {reasoning chain} `<answer_s>` {answer} | {question} |

Table 11: Our evaluation datasets. [†] indicates the development datasets we used to select fine-tuning hyperparameters.

| Task | Dataset name | Acronym | Metric | Score |
|---|---|---|---|---|
| Open-domain QA | MMLU (**?**) | MMLU | Acc. | nll |
| | Natural Questions (Kwiatkowski et al., 2019) | NQ | EM | nll |
| | TriviaQA (Joshi et al., 2017) | TQA | EM | nll |
| | [†]HotpotQA (Yang et al., 2018) | HoPo | EM | nll |
| | ELI5 (Fan et al., 2019) | ELI5 | Rouge-L | nll_token |
| Fact Checking | [†]FEVER (Thorne et al., 2018) | FEV | Acc. | nll |
| Entity Linking | [†]AIDA CoNLL-YAGO (Hoffart et al., 2011) | AIDA | Acc. | nll |
| Slot Filling | [†]Zero-Shot RE (Levy et al., 2017) | zsRE | Acc. | nll |
| | [†]T-REx (Elsahar et al., 2018) | T-REx | Acc. | nll |
| Dialogue | [†]Wizard of Wikipedia (Dinan et al., 2019) | WoW | F1 | nll_token |
| Commonsense Reasoning | BoolQ (Clark et al., 2019) | BoolQ | Acc. | nll_compl |
| | PIQA (Bisk et al., 2020) | PIQA | Acc. | nll_char |
| | SIQA (Sap et al., 2019) | SIQA | Acc. | nll_char |
| | HellaSwag (Zellers et al., 2019) | HellaSwag | Acc. | nll_char |
| | WinoGrande (Sakaguchi et al., 2019) | WinoGrande | Acc. | nll_char |
| | ARC-Easy (Clark et al., 2018) | ARC-E | Acc. | nll_char |
| | ARC-Challenge (Clark et al., 2018) | ARC-C | Acc. | nll_char |
| | OpenBookQA (Mihaylov et al., 2018) | OBQA | Acc. | nll_compl |

computational cost. For test set evaluation, we use the full set to ensure fair comparison with previous work. The language model instruction templates and retriever queries used in our evaluation are shown in Table 12. We randomly select few-shot examples from the official training splits of the KILT tasks, except for FEV, NQ and TQA, where we use the 64-shot examples released by Izacard et al. (2022b). For these three datasets, we also ensure that the 5-shot examples are subsets of the 64 examples. For retrieval augmented models, we use the top-1 relevant chunk to augment the prompt for each in-context few-shot example.

# E  ADDITIONAL EXPERIMENTS

## E.1  SCALING LAWS OF RETRIEVAL AUGMENTED LANGUAGE MODEL FINE-TUNING

We investigate the impact of the base language model size when retrieval-augmented instruction tuning is applied, and summarize the results in Figure 2. We combine the fine-tuned models with the base DRAGON+ retriever in this set of experiments.

Overall, all models substantially benefit from retrieval augmentation, with smaller models witnessing even bigger improvements. We further note that retrieval augmentation can be an effective strategy for enhancing the performance of smaller models (hence reducing pre-training and inference costs), given the 7B model leveraging $> 1$ retrieved chunks surpassed the performance of the vanilla 65B model on several tasks. This trend also differs across tasks. For tasks that primarily

Table 12: Language model prompts and retriever query templates used for our evaluation datasets. We did not perform retrieval for commonsense reasoning tasks evaluation.

| Task | LLM Prompt Template | Query Template |
|---|---|---|
| *Knowledge-Intensive Tasks* | | |
| MMLU | Background: {retrieved passage}\n\nQuestion: {question}\nA. {choice}\nB. {choice}\nC. {choice}\nD. {choice}\nA: {answer} | {question}\nA. {choice}\nB. {choice}\nC. {choice}\nD. {choice} |
| NQ, TQA, ELI5, HoPo, zsRE | Background: {retrieved passage}\n\nQ: {question}\nA: {answer} | {question} |
| AIDA | Background: {retrieved passage}\n\n{context}\nOutput the Wikipedia page title of the entity mentioned between [START_ENT] and [END_ENT] in the given text\nA: {answer} | {context} tokens between [START_ENT] and [END_ENT] |
| FEV | Background: {retrieved passage}\n\nIs this statement true? {statement} {answer} | {statement} |
| T-REx | Background: {retrieved passage}\n\n{entity_1} [SEP] {relation}\nA: {answer} | {entity_1} [SEP] {relation} |
| WoW | Background: {retrieved passage}\n\nQ: {turn_1}\nA: {turn_2}\nQ: {turn_3} ...\nA: {answer} | {turn_1} {turn_2} {turn_3} ... |
| *Commonsense Reasoning Tasks* | | |
| ARC-E, ARC-C | Question: {question}\nAnswer: {answer} | |
| BoolQ | {context}\nQuestion: {question}\nAnswer: {answer} | |
| HellaSwag | {context} {ending} | |
| OpenbookQA | {question} {answer} | |
| PIQA | Question: {question}\nAnswer: {answer} | |
| SIQA | {context} Q: {question} A: {answer} | |
| WinoGrande | {prefix} {answer} {suffix} | |

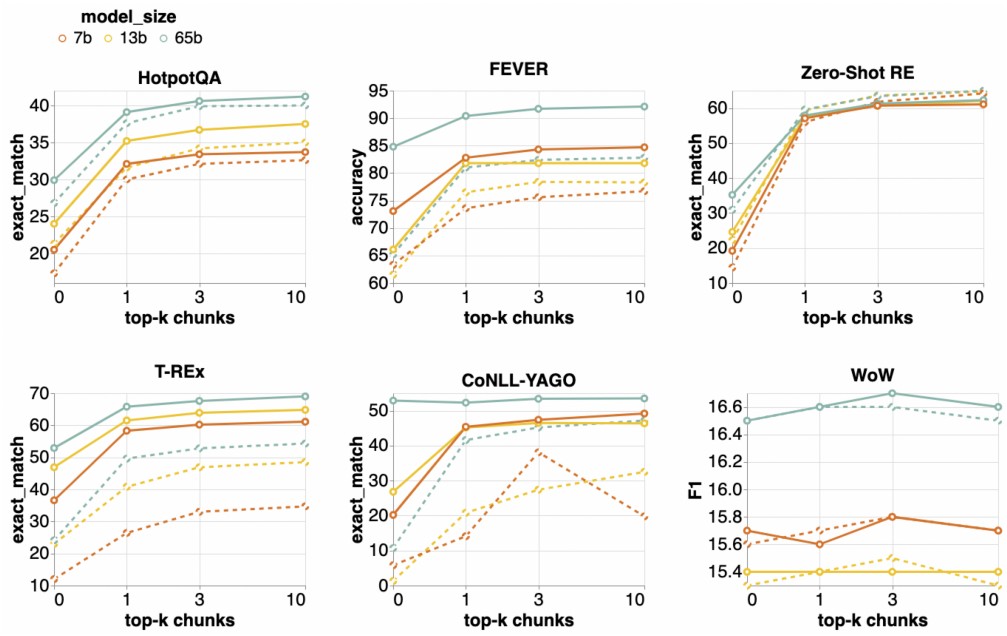

Figure 2: RA-IT model performance (combined with DRAGON+) across sizes 7B, 13B and 65B on our development tasks. 0-shot performance: dashed lines; 5-shot performance: solid lines.

measure one-hop fact look-up abilities (such as Zero-Shot RE and T-REx), retrieval augmentation provides significant improvements across all model sizes and can bring the performance of smaller models closer to that of their larger counterparts. For more complex tasks (such as HotpotQA and WoW), the advantage of using a larger LLM remains prominent.

Table 13: Comparison between parallel retrieval-augmentation and chunk concatenation. The results are obtained using the base DRAGON+ retriever.

| 0-shot | HoPo | FEV | AIDA | zsRE | T-REx | WoW | Avg |
|---|---|---|---|---|---|---|---|
| *top-3 chunks* | | | | | | | |
| RA-IT 65B (parallel) | 39.9 | 82.4 | 45.2 | 63.4 | 52.8 | 16.6 | 50.1 |
| RA-IT 65B (concat) | 39.5 | 83.9 | 52.2 | 65.2 | 47.9 | 16.6 | 50.9 |

### E.2 COMPARE PARALLEL RETRIEVAL AUGMENTATION TO CHUNK CONCATENATION

We adopt the parallel retrieval-augmentation approach proposed by Shi et al. (2023b) to reduce the prompt length, which is necessary in the few-shot settings (§2.1). However, this approach is computationally expensive when the individual prompts share long common prefixes (as in the few-shot setting). In addition, by separately encoding the text chunks, this approach is potentially less effective for knowledge synthesis compared to concatenating the retrieved text chunks in a single prompt. To understand the impact of using parallel retrieval-augmentation, we compare it to the chunk concatenation approach under the setting with top-3 retrieved text chunks. We conduct this experiment using the RA-IT 65B model and 0-shot evaluation,

According to Table 13, the two approaches perform closely on average with chunk concatenation demonstrating a small benefit. Specifically, parallel retrieval-augmentation under-performs chunk concatenation on FEVER and Zero-shot Relation Extraction, and perform on par on Wizard of Wikipedia. It also performs slightly better on HotpotQA, which is somewhat unexpected, given the dataset is specifically designed to necessitate multiple evidence sources for answering a question. We observe wider performance gaps between the two approaches on CoNLL-YAGO and T-REx, where concatenation performs much better on the former but worse on the latter.

It is worthnoting that the RA-IT 65B model has been fine-tuned using parallel retrieval augmentation, which potentially provides a benefit to using the same configuration during inference. We defer the investigation of fine-tuning with chunk concatenation to future studies. This direction appears promising, especially considering that state-of-the-art language models are progressively being trained with ever-larger context windows [17].

### E.3 RETRIEVAL CORPORA ABLATION

Table 14: Retriever settings: We report 5-shot dev set performance using LLAMA 65B and various retrievers in the REPLUG setting.

| 5-shot | MMLU | NQ | TQA | HoPo | FEV | AIDA | zsRE | T-REx | WoW | ELI5 | Avg |
|---|---|---|---|---|---|---|---|---|---|---|---|
| LLAMA 65B | 61.3 | 30.9 | 70.6 | 23.8 | 83.7 | 50.2 | 36.0 | 52.3 | 17.4 | 23.4 | 45.0 |
| *Retriever corpus ablation using* LLAMA *65B and the* DRAGON+ *retriever* | | | | | | | | | | | |
| CC only | 62.8 | 39.6 | 72.6 | 34.4 | 89.5 | 54.8 | 30.3 | 46.2 | 17.1 | 22.9 | 47.0 |
| Wiki 2021 + infobox | 62.2 | 42.0 | 71.2 | 41.8 | 89.8 | 62.2 | 65.3 | 73.1 | 17.7 | 22.2 | 54.8 |
| Wiki 2021 | 62.2 | 41.8 | 71.0 | 41.7 | 89.7 | 62.1 | 65.2 | 73.3 | 17.6 | 22.2 | 54.7 |
| Wiki 2018 | 61.5 | 42.6 | 70.7 | 40.4 | 90.8 | 62.1 | 51.3 | 59.8 | 17.6 | 22.5 | 51.9 |

Table 14 shows the impact of varying the retrieval corpora. In particular, we consider several subsets of our 399M retrieval corpus, namely CommonCrawl only (362M) and Wikipedia only (with and without infoboxes). We further compare with another Wikipedia snapshot (Wiki 2018) commonly used in the literature (Karpukhin et al., 2020). We observe that retrieving from Wikipedia only is beneficial for a number of KILT tasks such as AIDA and zsRE, as Wikipedia was the intended corpus for KILT tasks. We find that Wiki 2018 works better for NQ since the corpus is closer to the date of its data collection, similar to the observations by Izacard et al. (2022b). This indicates that our retrieval-augmented LM is faithful to the supplied retrieval corpus, and up-to-date information can be provided by updating the retrieval index at test time.

---

[17] https://openai.com/gpt-4

# F    EXAMPLES

In this section, we show the task prompts, the corresponding retrieved passages and model predictions generated by LLAMA 65B instruction-tuned with retrieval augmentation (RA-IT 65B) and LLAMA 65B instruction-tuned conventionally (IT 65B) on selected task examples.

## F.1    HOTPOTQA

We analyze the performance of the two models on the development set of HotpotQA in the zero-shot setting since under this setting RA-IT 65B outperforms IT 65B by a large margin. Table 15 show two examples from the HotpotQA development set where RA-IT 65B makes a correct prediction while IT 65B makes a wrong prediction. First, we observed that the dense retriever struggles to return useful text chunks for the multi-hop questoins in the HotpotQA dataset and most of the returned text chunks contains no information that helps the prediction. In this case, the IT 65B model shows a stronger tendency to be misled by distractors within the retrieved text chunk, since it has not been trained with noisy passages during fine-tuning. It also tend to predict "I don't know" more frequently[18], while the RA-IT 65B can ignore the noisy passages retrieved and predict the correct answer based on its parametric knowledge (Mallen et al., 2023). We also observe that in cases where both models generate wrong predictions because of the distractors (e.g. for the third text chunk in the second example), the generation probability of the wrong answer from RA-IT 65B is much lower; and in cases where both models ignore the noisy passages and rely on the parametric knowledge to make a prediction, RA-IT 65B outputs the correct answer with a higher probability (e.g. for the second text chunk in the first example).

---

[18]As discussed in §2.2, this behavior is induced by fine-tuning on SQuAD v2.0 (Rajpurkar et al., 2018), which trains the model to predict "I don't know" for passages that does not match with the given question.

Table 15: Example predictions in HotpotQA (dev set) in the 0-shot setting ensembling 10 retrieved text chunks. The top-3 retrieved chunks and the corresponding model predictions are shown. RA-IT 65B and IT 65B are used to generate these outputs.

| Prompt | $p_R$ | Output | | $\text{nll}_{LM}$ | |
|---|---|---|---|---|---|
| | | RA-IT | IT | RA-IT | IT |
| **Input:** Charlotte Hatherley initially came to prominence in a band formed in what year? **Label:** 1992. 
 **RA-IT 65B final prediction:** 1992 ✓ 
 **IT 65B final prediction:** 1997 ✗ | | | | | |
| Background: Charlotte Hatherley Born in London, Hatherley was brought up in West London and attended Chiswick Community School. Her music career began at the age of 15, when she joined British punk band Nightnurse. Two years later, with Ash looking for a guitarist to add to their live sound, Hatherley was hired after frontman Tim Wheeler saw her play at a Nightnurse gig. Hatherley's Ash debut was at Belfast's Limelight on 10 August 1997, and the following week the new lineup played the 1997 V Festival in front of 50,000 people. Her recording career with the band began later that year on the single Ä Life Less Ordinaryänd continued on the album Nu-Clear Sounds in 1998. Hatherley was a full-time member of Ash for eight years, playing on three studio albums, and wrote a handful of the band's songs, most notably G̈rey Will Fade, on the B-side of the single T̈here's a Star. The song was a cult favourite among fans, and eventually became the title track of Hatherley's debut solo album. On 20 January 2006 it was announced that Hatherley would be leaving Ash in an amicable breakup.\n\nQ: Charlotte Hatherley initially came to prominence in a band formed in what year?\nA: | 0.27 | 1992 | 1997 | 1.16 | 1.01 |
| Background: WM: Charlotte Hatherley only... so CD fans might still have to shell out big bucks for an import. Oh, in case you were wondering who Hatherley is, I first heard of her as the g̈irl guitaristïn the band Ash - a band that I have been a fan of since the early 90s when I was getting into all these Britpop-type bands. She naturally started doing her own solo material and left the band a few years ago. The last I heard of her was she was in the band new waver Client with Kate Holmes (not to be confused with the\n\nQ: Charlotte Hatherley initially came to prominence in a band formed in what year?\nA: | 0.21 | 1992 | 1992 | 0.46 | 0.98 |
| Background: Charlotte Hatherley Charlotte Franklin Hatherley (born 20 June 1979) is an English singer, songwriter, guitarist and soundtrack composer. She initially came to prominence as guitarist and backing vocalist for alternative rock band Ash. Since leaving Ash in 2006, she has pursued a solo career and acted as a touring instrumentalist for Bryan Ferry, KT Tunstall, Bat for Lashes, Cold Specks, Rosie Lowe and Birdy. Hatherley has also been a touring member of NZCA Lines and is currently musical director for South African artist Nakhane.\n\nQ: Charlotte Hatherley initially came to prominence in a band formed in what year?\nA: | 0.13 | 1992 | I don't know. | 0.54 | 0.72 |
| **Input:** Oxley Highway ends at a coastal town that had how many inhabitants in June 2016 ? **Label:** 45,698. 
 **RA-IT 65B final prediction:** 45,698 ✓ 
 **IT 65B final prediction:** I don't know. ✗ | | | | | |
| Background: Oxley Electorate: Ipswich Motorway: 1 Dec 2016: House debates (OpenAustralia.org) Oxley Electorate: Ipswich Motorway The Ipswich Motorway is a vital link supporting the Queensland economy. It forms part of the national land freight network providing connectivity for industry to the Acacia Ridge intermodal facility, the major industrial area of Wakool and the Brisbane markets at Rocklea2̌014in the member for Morton's electorate2̌014which are the state's largest fruit and vegetable markets and a major centre for produce on the east coast. The section of the motorway is over capacity with 93,000 vehicles on average each day, including up to 12,000 freight vehicles. Numbers are increasing each year at an average of four\n\nQ: Oxley Highway ends at a coastal town that had how many inhabitants in June 2016 ?\nA: | 0.25 | 10,000 | I don't know. | 7.27 | 0.61 |
| Background: Post Offices For Sale NSW — Lotto — Newsagencies — Marlow & Co South Wales about 390 km north of Sydney, and 570 km south of Brisbane. The town is located on the Tasman Sea coast, at the mouth of the Hastings River, and at the eastern end of the Oxley Highway. The town with its suburbs had a population of 45,698 in June 2016. Port Macquarie is a retirement destination, known for its extensive beaches and waterways. Port Macquarie has a humid sub-tropical climate with warm, humid summers and mild winters, with frequent rainfall spread throughout the year. Port Macquarie2̌019s central business district contains two shopping centres, a marina, the beginnings of\n\nQ: Oxley Highway ends at a coastal town that had how many inhabitants in June 2016 ?\nA: | 0.15 | 45,698 | 45,698 | 0.18 | 0.38 |
| Background: The Long Paddock - THE LONG PADDOCK The Long Paddock 4x4, 4WD, caravan, camper trailer, camping products reviews, tests, comparisons by Mark Allen The Long Paddock west, the Oxley Highway is the track you2̌019ll be aiming for and Tamworth is the major western town of reference on the map. Once you2̌019re in the main streets of Port, you2̌019ll wonder no more why in excess of 76,000 people now call the area home. As a rough breakdown, the majority of locals are 25 to 44, followed closely by the 45 to 64 year old bracket 2̌013 just perfect for all you thrill seeking middle aged folk and laid back grey nomads and let2̌019s not forget about the younger set that now have oodles of schooling and after-schooling\n\nQ: Oxley Highway ends at a coastal town that had how many inhabitants in June 2016 ?\nA: | 0.12 | 76,000 | 76,000 | 4.85 | 0.93 |

