# OpenReview forum: "RA-DIT: Retrieval-Augmented Dual Instruction Tuning"
_ICLR.cc/2024/Conference — ICLR 2024 poster_

### Official Review · Reviewer_i49b · 2023-10-29

**Soundness:** 2 fair
**Presentation:** 2 fair
**Contribution:** 2 fair
**Rating:** 5
**Confidence:** 3

**Summary:**

This paper introduces a two-stage fine-tuning process aimed at enhancing a pre-trained language model's performance. In the first stage, the focus is on improving the LM's ability to effectively utilize retrieved information, while in the second stage, the retriever is fine-tuned to provide more contextually relevant results as desired by the LM. The study reveals that both of these stages contribute significantly to performance enhancements. Furthermore, when both stages are combined, even greater improvements are observed. The proposed model, RA-DIT 65B, achieves state-of-the-art results in various knowledge-intensive zero-/few-shot learning tasks, surpassing existing in-context RALM approaches by a substantial margin.

**Strengths:**

* One of the standout strengths of this paper is its ability to outperform other presented baseline models.

* The paper's approach offers practical value by demonstrating that the fine-tuning process is lightweight. This means it can be implemented efficiently, making it a more accessible and feasible solution for real-world applications.

* The method presented in the paper is well-motivated and clearly described, enhancing its accessibility and potential for replication. This transparency in methodology ensures that other researchers can easily understand and build upon the work, further advancing the field.

**Weaknesses:**

* The paper's architecture lacks significant novelty as it primarily relies on existing models and introduces minor modifications. This may limit its impact and originality in the field.

* The paper raises concerns about the accuracy of the ATLAS scores in the 64-shot fine-tuning results presented in Table 2. Discrepancies between the reported scores and those from the ATLAS paper, as observed in Table 10 of the ATLAS paper, undermine the credibility of the findings.

* The paper does not include a comparison with a strong baseline model, such as FID+RS (Hofstätter et al., 2022). If included, it is suggested that FID+RS would outperform the proposed model, based on reported scores from the FID+RS paper. Additionally, FID+RS has the advantage of a significantly smaller model size, which raises questions about the efficiency and resource requirements of the proposed model. (11B vs 65B)

**Questions:**

If any of the concerns raised stem from misunderstandings, kindly bring them to my attention. I would greatly appreciate the opportunity to gain a better understanding and, if necessary, reevaluate the content in question

---

> ### Author Response · Authors · 2023-11-11
> **Evaluation Setting and Performance Comparison Clarification**
>
> Thanks for your time and feedback. We would like to clarify misunderstandings regarding our approach and result comparison based on the weaknesses brought up.
>
> 1. **In our paper, evaluations on NQ and TQA were conducted using the original data released by Kwiatkowski et al. (2019) and Joshi et al. (2017). Correspondingly we quoted the 64-shot results from Table 8 in the ATLAS paper for comparison.** The NQ and TQA results in Table 10 of the ATLAS paper are based on the customized evaluation sets released by the KILT Benchmark (Petroni et al. 2021), which are not comparable to ours. (Note that for the other 6 KILT tasks, we quote the numbers from Table 10 of ATLAS paper for comparison.)
> 2. Thanks for bringing Hofstätter et al. (2022) to our attention; we will cite the paper in our related work section. However, it is important to note that **their evaluation setting differs significantly from ours**. **Our paper primarily evaluates the fine-tuned RALM in the few-shot setting, whereas Hofstätter et al. (2022) employ a multi-task supervised-learning setting**, leading to their higher reported performance. Specifically, the fine-tuning datasets we used, as detailed in Table 1, do not overlap with our evaluation datasets; and our main experiments are conducted in the in-context few-shot learning setting (Brown et al. 2020). In contrast, Hofstätter et al. (2022) train their model using a re-balanced combination of training sets from the KILT task (which comprises approximately 808K examples) and evaluate the model on the same set of tasks.

---

> > ### Comment · Reviewer_i49b · 2023-11-22
> > **Acknowledging Clarification**
> >
> > I appreciate the clarification provided. Your comments are sound and well-founded, prompting me to revise my rating from 3 to 5.

---

> ### Author Response · Authors · 2023-11-20
>
> Thank you again for your valuable feedback! We would like to politely follow up and inquire whether our clarification has addressed your concerns. If you have any follow-up questions, please let us know and we would be more than happy to answer them. Thanks!

---

> ### Author Response · Authors · 2023-11-22
> **Re: Architecture Novelty Concerns**
>
> Thank you for your response and revising the rating accordingly. We would like to respectfully follow up and see if there are any remaining concerns we can address for a further improved rating.
>
> > The paper's architecture lacks significant novelty as it primarily relies on existing models and introduces minor modifications. This may limit its impact and originality in the field.
>
> Regarding the weakness raised on architecture novelty, we highlight that fusing independently optimized LMs and retrievers through lightweight fine-tuning is a strength of our approach **with potentially greater practical value and impact**, as also noted by Reviewer 68qv. This strategy extensively leverages existing pre-training efforts in both the LM and retriever components, which are typically resource-intensive and feasible to carry out in only a limited number of organizations.
>
> Additionally, we demonstrate that **the implementation complexity and computation costs for fine-tuning in-context RALM architectures can be further reduced through the dual fine-tuning framework proposed**, which does not involve end-to-end gradient back-propagation across both components **while still maintaining effectiveness**.

---

### Official Review · Reviewer_HttW · 2023-11-01

**Soundness:** 3 good
**Presentation:** 4 excellent
**Contribution:** 3 good
**Rating:** 6
**Confidence:** 3

**Summary:**

The paper proposed the Retriever-Agument dual Instruction tuning, aimed to get the pretrained language model to better understand the retrieved text. The paper constructs a dataset, consisting of different domains (Open book QA, summarization, general conversation, etc.). The dataset improves the performance of the original model by a good margin.

**Strengths:**

1. The paper presents very solid experiments on the instruction tuning.
2. The paper is very well presented.
3. The ablation of the paper is solid.

**Weaknesses:**

1. The author mentioned the early stopping, training the model for ~500 steps would bring the best performance. How is the dataset size scaling with this? Can we change the cap of 7000 data points per domain to smaller or larger? will the performance change?
2. Is there a criteria of how to choose the domains? And does jointly training on multiple domains help boost the performance? The choice of different domains seems a bit random.
3. How sensitive is the model to different type of retrievers? Like can contriever/dpr work in this setting?

**Questions:**

Please see above.

---

> ### Author Response · Authors · 2023-11-20
> **Response to Review Questions**
>
> Thanks for your time and feedback. Please find our responses to the questions below:
>
> 1. When early stopping is used, the total amount of data seen by the model is equal to the product of batch size and the number of steps taken, and the model may not be seeing a full epoch of data before fine-tuning stops. As outlined in Appendix B, our implementation does not apply a hard cutoff of 7500 examples per dataset. Instead, we recompute the dataset distribution given this cap and sample from all datasets based on the newly adjusted distribution. We chose the cap of 7500 examples after observing the fine-tuned model substantially outperformed the baseline model on our development sets. However, we avoided carefully tuning this hyperparameter to prevent overfitting to the development sets. **We have updated Appendix B to reflect the decision-making process**.
>
> 2. Prior work (Chung et al. 2022; Iyer et al. 2022) have demonstrated that fine-tuning the language model on a diverse collection of instruction-based datasets leads to improved model generalization for unseen instructions. We hence adopt a similar strategy by combining five categories of fine-tuning tasks to enhance the language model's knowledge utilization (dialogue, open-domain QA, chain-of-thought reasoning) and to improve its contextual awareness for prediction generation (reading comprehension, summarization). These categories were selected given their representativeness of practical knowledge-intensive language tasks. **We have updated Appendix B to explain this choice**.
>
> 3. Please refer to our ablation study in Table 7, where we evaluate our model's performance using various retrievers, including DRAGON+ (our default choice), Contriever, and a version of Contriever fine-tuned with the MS-MARCO dataset (Izacard et al. 2022). We found the model performance is indeed sensitive to the effectiveness of the retriever. Notably, using DRAGON+ results in significant performance improvement compared to using both versions of Contriever.

---

> > ### Author Response · Authors · 2023-11-22
> >
> > Thank you again for your valuable feedback. As we approach the end of the discussion period, we want to check in and inquire whether our previous response has addressed your concerns. If you have any follow-up questions, or any concerns we haven't addressed yet, please let us know and we would be more than happy to answer them.

---

### Official Review · Reviewer_5vF7 · 2023-11-04

**Soundness:** 2 fair
**Presentation:** 2 fair
**Contribution:** 3 good
**Rating:** 6
**Confidence:** 4

**Summary:**

This paper proposes retrieval-augmented instruction tuning for pre-trained LLM. It also finetunes retriever.

**Strengths:**

- Important topic.
- Overall good results on knowledge intensive tasks in contrast to previous methods.

**Weaknesses:**

- Fine-tuning the retrievers seems to be unnecessary given the very marginal improvement. However, this paper is written in a way that encourages the fine-tuning of retrievers.
- The retrieval-augmented instruction tuning may encourage hallucination. See detailed comments.

**Questions:**

Detailed comments:

1. “To stay within the context window size limit, each retrieved chunk is prepended individually to the prompt”
- Llama has 2k context. What’s the length of the retrieved chunk? How many chunks (top-?) are used in the experiment? Did you try to pack top-5 all-together into the prompt?

2. “Secondly, even state-of-the-art retrievers can falter and return inaccurate results. By training the LLM to make correct predictions when a wrong retrieved chunk is given, we enable the LLM to ignore misleading retrieval content and lean into its parametric knowledge in such cases”
- This process may encourage the LLM to hallucinate if i) answer is not in retrieved context, and ii) the required knowledge for answering the question is not stored in LLM parameters.

3. For baseline Llama 65B, did you apply instruction tuning using the same blended dataset? In Table 4, the improvement from IT 65B vs.RA-IT 65B is pretty marginal with the top-1 chunk.

4. Could the authors also provide the results using top-5 in Table 4?

5. In Table 2 main results, why is Llama 65B so bad on NQ? Do you evaluate it in a close-book manner i.e., non-retrieval setting?

6. In Table 5, fine-tuning retriever seems to only provide very marginal improvements.

Concurrent work:

InstructRetro: Instruction Tuning post Retrieval-Augmented Pretraining.
It would be recommended to include relevant discussion in related work.

---

> ### Author Response · Authors · 2023-11-20
> **Response to Review Questions**
>
> Thanks for your time and feedback. Please find our responses to the questions below:
>
> 1. Our retrieved chunk has a maximum length of 200 words (sec 3.1). We use up to 10 chunks in the experiment. For 5-shot in-context learning, we have to adopt the parallel augmentation approach (Shi et al. 2023) instead of concatenating all retrieved chunks to stay within the 2k context window of Llama. For 0-shot experiments, we adopt the same setting for consistency. **We have added the comparison of parallel retrieval-augmentation vs. concatenation using top-3 chunks in the 0-shot setting in Appendix E.2.**
>
> 2. We agree this hypothesis might be valid; however, other methods including the base LM would likewise be ineffective when condition i) and ii) simultaneously hold. Nevertheless, as highlighted in footnote 2 of our paper, by incorporating the SQuAD 2.0 dataset – a reading comprehension dataset that includes questions that cannot be answered with the given support documents – we encouraged the model to respond with “I don’t know” when it lacks sufficient information to provide an answer instead of hallucinating one. This approach could actually reduce hallucination but we leave a careful study to future research, where the focus could be on enabling the model to abstain from responding or to request additional information when needed.
>
> 3. Yes, the IT 65B baseline is instruction-tuned with the same blended dataset without retrieval augmentation. We hypothesize that the competitiveness of the IT 65B model in the top-1 setting is due to the inclusion of reading comprehension and summarization tasks in our fine-tuning dataset blend. Such tasks likely improve the model's capacity for contextual interpretation, which is helpful for generating accurate answers. However, **without retrieval-augmentation, the IT 65B model is not exposed to the broader and potentially noisier content returned by the retriever. This limits its ability to generalize, especially when more retrieved chunks are included.**
>
> 4. **We have added the top-3 results in Table 4.** For both Llama 65B and RA-IT 65B, we observe monotone performance increase as the number of retrieved chunks increase. **The performance of IT 65B slightly dropped going from top-3 to top-10, further validated that this baseline is less effective in processing noisier context.**
>
> 5. In Table 2 main results, the NQ results of Llama 65B are obtained using the close-book setting. Llama 65B performs poorly on NQ in the 0-shot setting because it struggles to generate answers that match the short format of the ground truth responses.
>
> 6. In Section 5.2, we presented an ablation on the dual instruction tuning approach where we analyze the individual impact of LM-ft and R-ft. As shown in Table 6, R-ft consistently improves the performance of both the base Llama 65B and the RA-IT-ed Llama 65B, Our final approach combines LM-ft and R-ft since it yields further gains over using either LM-ft or R-ft alone.
>
> 7. InstructRetro (Wang et al. 2023) is a very interesting concurrent work which applies instruction tuning to the Retro architecture (Borgeaud et al. 2021). **The paper was released on ArXiv post ICLR submission**, which reinforces that enabling LMs to better integrate external knowledge in downstream tasks is an important direction. This approach involves encoding and integrating retrieved content into the language model (LM) using separate encoders and cross-attention mechanism. In comparison, RA-DIT adopts a more streamlined architecture and training procedure by separating the fine-tuning processes of the LM and the retriever. Although the differences in base LM, fine-tuning datasets and inference settings make direct comparisons between the two models challenging, RA-DIT 65B compares favorably to InstructRetro 48B in zero-shot settings on shared evaluation datasets: TriviaQA (RA-DIT 75.4 vs. InstructRetro 65.6) and NQ (RA-DIT 35.2 vs. InstructRetro 38.9). We will add this discussion to our related work.

---

> > ### Author Response · Authors · 2023-11-22
> >
> > Thank you again for your valuable feedback. As we approach the end of the discussion period, we want to check in and inquire whether our previous response has addressed your concerns. If you have any follow-up questions, or any concerns we haven't addressed yet, please let us know and we would be more than happy to answer them.

---

### Official Review · Reviewer_68qv · 2023-11-05

**Soundness:** 4 excellent
**Presentation:** 3 good
**Contribution:** 4 excellent
**Rating:** 8
**Confidence:** 4

**Summary:**

This paper proposes a method to endow LLMs with retrieval-augmented generation capabilities through a lightweight fine-tuning procedure that is in-between costly ad-hoc retrieval-aware pretraining and cheaper but subobtimal post-hoc integration techniques.
The proposed approach consists in two fine-tuning steps: one to update a pretrained LLM to make good use of retrieve information and one to fine-tune the retriever itself.
The paper shows that each of these procedures improve the performance on tasks that require knowledge utilization and contextual awareness, achieving state-of-the art performance across multiple knowledge-intensive benchmarks.

**Strengths:**

* The method is well-motivated in terms of practical applicability in providing pretrained LLMs with retrieval capabilities
* Reproducibility is ensured by the detailed documentation of the used instruction tuning datasets and at what stage of fine tuning they are being used
* Baseline comparisons and ablation studies are thorough and convincing

**Weaknesses:**

* The proposed components are well motivated and clear, but the presentation could make it easier to get a sense of the paper in a quick glance. For instance, it might be beneficial to expand Figure 1 and its caption to help in that.
* The method is presented as two independent fine-tuning steps. On the other hand, the two fine-tuned elements depend on each other: LM-fine-tuning uses the retriever, while the retriever fine-tuning needs the LM to score the outputs. It could be beneficial for the understanding of the paper to emphasize this point.

**Questions:**

* Connecting back to the point above, it seems that the final version of the algorithm consists in fine-tuning the LM using the pretrained retriever, and then the retriever is fine-tuned using this fine-tuned LM. Is that indeed the case?
* Either way, what would be the drop in performance in actually performing the two fine-tuning steps independently, as opposed to fine-tuning the retriever with the (as said presumably) already fine-tuned LM?

---

> ### Author Response · Authors · 2023-11-13
> **Fine-tuning Process Clarification**
>
> Thanks for your time and feedback. In our paper, we fine-tune the retriever using the base LM (as opposed to the already fine-tuned LM); and similarly, we fine-tune the LM using the retrieval results from the base retriever. The term “independent” in our paper is used to indicate that the LM fine-tuning step and retriever fine-tuning step can be carried out concurrently. We hope this helps to clarify the depiction in Figure 1, which illustrates the non-sequential nature of these two steps. As shown in Table 6, we found combining the separately fine-tuned LM and retriever yields further gains. While we leave sequential fine-tuning out of the scope of this work, we recognize it as a conceptually compelling alternative method that merits exploration in future research.

---

> > ### Comment · Reviewer_68qv · 2023-11-21
> >
> > Thank you very much for the clarification. I don't know how I missed the the two fine-tuning steps are carried out concurrently. I think this makes the proposed method even more convincing and of practical interest. Even though sequential fine-tuning might turn out to be better, it is quite remarkable that the algorithm works robustly by fine-tuning the retriever and the LLM concurrently.

---

### Author Response · Authors · 2023-11-23
**Author Response Summary**

We thank all reviewers for their feedback and helpful suggestions. We appreciate the positive comments (e.g. *"method is well-motivated"*, *"outperform other presented baseline models"*, *"solid experiments on the instruction tuning"*, *"detailed documentation of the used instruction tuning datasets"*, *"approach offers practical value and of practical insterest"*) as well as the constructive criticism. We have revised our paper draft accordingly, and included pointers to each specific change in the individual responses below. Revisions in related work are also highlighted in blue in the PDF.

---

### Meta-Review · Area_Chair_L9Va · 2023-12-17

**Metareview:**

The paper provides a simple method to fine-tune LLMs to give them retrieval-augmented generation capabilities. The method is clearly explained, well-motivated and empirical results are extensive and thorough. Most reviewers recommended acceptance and all the concerns raised seem to be well-addressed by the authors. We encourage the authors to reflect all the reviewer feedback in the updated version.

**Justification For Why Not Higher Score:**

I do not have anything negative that stands out in the paper - it mostly feels a little straightforward and that's why I am hesitant to recommend for a spotlight (reviewer score and sentiment also seems to match this).

**Justification For Why Not Lower Score:**

The method works well, baselines are correctly compared to, and the paper is generally well written

---

### Decision · Program_Chairs · 2024-01-16

Accept (poster)